# Brief Theoretical Overview of Bi-Fe-O Based Thin Films

**DOI:** 10.3390/ma15248719

**Published:** 2022-12-07

**Authors:** Denis Misiurev, Pavel Kaspar, Vladimír Holcman

**Affiliations:** Department of Physics, Faculty of Electrical Engineering and Communication, Brno University of Technology, Technicka 2848/8, 61600 Brno, Czech Republic

**Keywords:** Bismuth ferrite, multiferroic, thin film, ferroelectric, magnetic, PDL, ALD, leakage current, hysteresis

## Abstract

This paper will provide a brief overview of the unique multiferroic material Bismuth ferrite (BFO). Considering that Bismuth ferrite is a unique material which possesses both ferroelectric and magnetic properties at room temperature, the uniqueness of Bismuth ferrite material will be discussed. Fundamental properties of the material including electrical and ferromagnetic properties also will be mentioned in this paper. Electrical properties include characterization of basic parameters considering the electrical resistivity and leakage current. Ferromagnetic properties involve the description of magnetic hysteresis characterization. Bismuth ferrite can be fabricated in a different form. The common forms will be mentioned and include powder, thin films and nanostructures. The most popular method of producing thin films based on BFO materials will be described and compared. Finally, the perspectives and potential applications of the material will be highlighted.

## 1. Introduction

The current trends require more minimalistic technologies and higher energy efficiency which leads to an increasing interest of developing new types of materials. Recent trends of material science require invention of new materials which would effectively combine multiple properties together with temperature and chemical stability. New and popular trends of designing modern devices involve control of both magnetic and electrical parts, thus creating multifunctional devices with a combination of ferroelectric and ferromagnetic effects. This combination offers perspective opportunities of designing new and complex microelectronic systems. The materials, which combine both ferroelectric and magnetic properties, are extremely rare and vital for solving a wide variety of problems and are implemented in many applications.

Multiferroic materials stand out, since their electrical and magnetic properties can be changed by the electromagnetic field. Multiferroic materials aim to solve the problem related to creating more energy–efficient microelectronics and applications with faster speed reaction, telecommunication, etc. 

The actual definition of the word multiferroic stands for combination of ferroelectric and ferromagnetic effect of a material. The current definition of multiferroic also includes the antiferroelectric effect. Nowadays, multiferroic materials draw a lot of attention as they can be implemented in a wide variety of new devices and detectors. Since multiferroic materials exhibit ferroelectric and magnetic properties, they became an essential part of many nanostructures. These materials also require less power than conventional systems to operate.

A unique advantage of single phase multiferroic based thin film materials is the fact that they suggest appealing and prosperous ways of creating new materials with a unique combination of different substrates which emphasize the advantages of individual materials. Single phase multiferroic materials which exhibit strong magnetic coupling are rare. Therefore, bismuth ferrite is the only intensively studied material.

BFO is a multiferroic material with great potential. BFO is an outstanding material which possesses ferroelectric and magnetic properties simultaneously. The material has a significant advantage in terms of photovoltaic devices due to a narrow bandgap, suppression of recombination of electron–hole pair and wide absorption range. Manipulation of electromagnetic coupling by electric or magnetic field makes BFO-based materials potential candidates for spintronic applications. 

Thin films of BFO material were and still are very popular for the development of modern nanomaterials. Thin films allow numerous modifications of morphology and thicknesses of the final product. In comparison to BFO ceramics, thin films can be produced in a relatively low range of operational temperatures, which enhance the overall performance of the final film. BFO–based thin films were widely used in ferroelectric memory storage devices due to high remanent polarization. In addition to memory devices, thin films play an important role in miniature antennas and microwave MEMS devices. 

Over the past decade, BFO has been intensively used in BFO–based ceramics due to its high temperature resistance and the coexistence of para–ferro electric phases. In addition to the morphotropic phase boundary, the perovskite chemical structure of BFO material allows implementation of impurities, thus enhancing electromagnetic coupling. Another considerable advantage of BFO material is manipulation with its ferrotoroidic system by the electric/magnetic field. These properties combine with the high conductivity of BFO and make it a promising material for the development of BFO–based ceramics.

Although an outstanding material with unique properties among other materials, there are still challenges that need to be addressed in the near future. It is necessary to find potential solutions to decrease high parasitic current and increase relatively low electromagnetic coupling for implementation into large-scale devices. The physical mechanism of photovoltaic response is still unknown and requires further and deeper investigation. 

## 2. Characterization of Bismuth Ferrite Material

BFO material is one of a few multiferroic materials which shows a simultaneous ferroelectric and ferromagnetic effect [1]. The material was intensively studied over the past decade because of BFO versatile multifunctionality. It was proven that BFO is the only material which has the strongest electromagnetic coupling at room temperature range among all potential materials making it an excellent candidate [2] for ferroelectric sensors. 

Considering all multiferroic materials, thin films based on BFO have been successfully used for microelectronic and optoelectronic devices due to low band gap. The principal of ferroelectricity at room temperature is based on a lone–pair mechanism, where valence electrons of Bi^+3^ create a localized magnetic dipole and are involved in SP–hybridization [3]. The local magnetic dipole leads to the creation of spontaneous (remnant Pr [3]) polarization around 90–100 (μCcm^−2^) [4,5] which is a main reason BFO is an exceptional multiferroic material. 

Magnetoelectric materials belong to multiferroic materials with a correlation of magnetic and ferroelectric properties. Magnetic properties (magnetization) and electric properties (polarization) can be controlled by the electric and magnetic field. The material is synthesized into single and multi–phase molecules. 

BFO is popular not only due to its thermal stability but also due to its high range of polarization (~90–100 μCcm^−2^) [4,5].

### 2.1. Structural Characterization of BFO Material

The most common crystallization form of Bismuth ferrite material is a crystallographic structure represented by a symmetric rhombohedral (Figure 1) [2,3,6] centrally orientated perovskite [6] structure where parameters a, b are lattice parameters and c is a hexagonal parameter. The typical ferromagnetic perovskite parameters of BFO material stand within a = b = (5.7–6.7) Å [7,8] and hexagonal c = (13–14) Å [5,9]. The angle α is around 60 degrees [10]. The space group orientation is R3c. The parameters of the cubic rhombohedral structure are a = b = c ~4 Å and angle α stays roughly under 90 degrees.

R3c space orientation stands for rhombohedral symmetry group, where the direction of one symmetry is c, and the other direction of symmetry being perpendicular to c. Rhombohedral order is represented by two symmetry directions, which are c and b. The coordinate system for the rhombohedral group corresponds to [9] with lattice parameters a = b = c and α ≠ 90°.

The typical Perovskite structure (Figure 2) includes an FeO unit which is inserted into the Bi cubic rhombohedral crystal structure. Oxygen anions and Fe cations create octahedral orientation closed by Bi ions.

One of the most significant disadvantages which limits usability of BFO [6] is high parasitic current (low electrical resistance). High parasitic current accrues in the BFO composite due to defects related to the secondary phase of BFO such as absence of Bi or oxygen atoms during the preparation process. There are many reasons which cause high parasitic current; however, the primary cause is a high amount of impurities during the deposition process, and the chemistry of the BFO unit. One potential defect is related to the evaporation of bismuth during the preparation process, thus changing the chemical valence of Fe^+2^ to Fe^+3^ [11,12] ions due to the evaporated Bi^+2^ [11,12] cations and oxygen vacancies. Uncompensated vacancies lead to a distribution of spin moment of the entire cell unit.

The absence of oxygen anions to compensate the charge of lower chemical valency of Fe^+2^ causes a significant decrease of electrical resistance, and thus an increasing leakage current. 

The parasitic current becomes more prominent in multi–phase composites (Bi_2_Fe_4_O_9_, BiFeO_3_ etc.), thus it is becoming more popular to find methods to reduce parasitic current. To lower the leakage current, the structure of the composite has to be changed by adding a small amount of impurities [13]. Some papers prove a significant decrease in leakage current by implementing rare–earth materials [6] such as titanium, chromium and manganese. The leakage current [14,15] causes a huge limitation of implementation of BFO material into complex sensor devices.

### 2.2. Magnetic Properties of BFO Material

The crystallographic structure of BFO [14] is represented by a rhombohedral centrally orientated perovskite structure [15]. Oxygen anions create an octahedral formation in the crystal, thanks to these bands the system showing a nonzero [8] ferromagnetic response. Ferroelectric properties of the material are highly dependent on these oxygen bands [4,7]. Bismuth atoms create a cubic rhombohedral structure next to octahedral units of FeO, which is located inside of rhombohedral structure. Spin interaction of bismuth atoms and the resulted spin moment of the octahedral FeO unit leads to the formation of ferroelectric response. Spin moment of the Bi^+3^ electron pair (6s^2^) [16] shares an electron pair with residual moment of Fe^+3^ caused by a weak magnetic response [16]—G–type magnetization order (Figure 3). 

Magnetization order characterizes the orientation of magnetic couples and angular moment of atoms. In G–type magnetization order, all nearest magnetic dipoles are oriented antiparallelly (Figure 3), causing antimagnetic distribution. Recent studies show that magnetic properties of the rhombohedral structure of BFO are size-dependent (Figure 4) [5,6]. It was proven that the ferroelectric could be greatly enhanced by changing the size of BFO [14] particles. As BFO crystals are smaller, the magnetic response is stronger due to a closer and stronger interaction of magnetic dipoles [5]. Ferroelectric response is stronger when the size of particles is smaller due to the lower interaction of Bi and Fe spin moments, resulting in overall enhanced ferroelectric response. Critical thickness, in which the antiferromagnetic response dominates over ferroelectric response, starts around 150 nm [17]. 

The overall magnetic spin moment of the rhombohedral structure is perpendicular [7] to its central axis which leads to the existence of small magnetization ability. The magnification is affected by oxygen bands in the same way as ferroelectric properties. 

It is necessary to mention that ferromagnetic properties start to occur by the overall suppression of cycloid magnetic moment [18], otherwise ferroelectric and magnetic properties are subdued. The cycloid moment can be broken under certain conditions: by implementing the magnetic field, magnetization, chemical additives, and strain into thin films or heterostructures the cycloid moment is suppressed, allowing ferroelectric and magnetic properties to occur [18]. 

The existence of cycloid spin moment [18] is given by flexomagnetoelectric interaction. Considering flexomagnetoelectric [18] interaction, the electric polarization is offset by spin modulation. It was shown that magnetic properties become negligible if the size of nanoparticles is greater than 65 nanometers [8].

Another significant advantage of BFO is the fact that BFO has outstanding temperature stability (Figure 5 and Figure 6). The temperature, where it changes from antiferromagnetic to permanent magnetic material, is also known as Neel temperature (Figure 6). The Neel [7] temperature of BFO stays around of 320 to 350 °C [6,7]. The material is capable of withstanding a high range of temperatures and retain its magnetic properties (Curie Temperature is 850 °C [6,19]). Considering the outstanding thermal stability of the material, including high Curie and Neel temperatures, the material shows little change with temperature differences.

In addition to ferromagnetic properties, BFO also shows massive spontaneous magnetization [20].

It has always been a trend to find a way to enhance the magnetic and ferroelectric properties of BFO material. Since magnetic and ferroelectric properties are greatly affected by oxygen bands, adding additional FeO will greatly increase the magnetic properties [6].

Even while possessing the strongest ferromagnetic response of multiferroic properties, the multiferroic properties of BFO material remain weak to be implemented into large-scale devices. In addition, molecules of BFO are exposed to aggressive oxidation of the oxygen atmosphere. It has been a hot topic of many discussions to figure out a potential solution to decrease the parasitic current and enhance the multiferroic response. The weak ferroelectric and magnetic response (antiferromagnetic response) for large-scale devices has been a significant drawback of the material for wide implementation (Figure 7). However, in the year 2003, Ramesh developed artificial heterostructures with significantly higher multiferroic properties [21].

## 3. Thin Films of Bismuth Ferrite 

Bismuth ferrite [14] has been under intense study because of the strongest and unique combination of magnetic and ferroelectric properties. In addition to unique multiferroic properties, BFO demonstrate a photoelectric, weak piezoelectric response and strong dielectric properties, which could be enhanced by adding additional impurities of rare–earth materials. 

### 3.1. Morphology and Composition of Produced Thin Film

Thin films have drawn a lot of attention and popularity to be implemented into many prosperous applications and complex microelectronic devices [22]. Thin films of BFO are extremely popular nowadays due to the extreme degree of versatility [12,23]. Design of thin films [24] offers numerous modifications of morphology (Figure 8B), particle size (Figure 8A) and overall chemical compound of produced films. The versatility of thin films is achieved by various deposition methods. 

BFO thin films demonstrate different crystallographic orientations based on ration of lattice parameters with various orientation, with ration angles being 71°, 109° and 180°, the operation temperature during the deposition process and substrate morphology. Thin film can be synthesized into rhombohedral, relaxed rhombohedral (bulk), tetragonal and strained tetragonal phases. There is a slight change in the crystallographic structure of substrate structure to bulk phase with increasing film thickness caused by strain relaxation. This change effects the symmetry of the resulted BFO film [26]. 

As was mentioned previously, to enhance the performance of BFO material, it is necessary to suppress cycloid moment. According to fundamental studies [27,28] thin films with particles smaller than 62 nm have a significant increase in ferromagnetic response. Increased performance of BFO thin films can be described as follows: particle size stays within the range of 62 nm leads to the modulation of spiral spin moments resulting in greater enhancement of ferromagnetic properties [27]. With even smaller particle size, BFO material demonstrates a negligible ferromagnetic response due to insufficient compensation of spin moments. Uncompensated moments result in a change of bond angle and bond length of tetragonal and decrease of tilt of octahedral FeO units. Corresponding to the first principal calculation, the tetragonal bond angle is crucial for the transformation between ferromagnetic and antiferromagnetic phases. The change of bond aFngle suggests there is a thickness dependence of thin films similar to bulk.

The easiest and comprehensible way of offset spiral spin moment is by straining BFO material into thin films (Figure 9). It has been found, that under optimal strain coefficient (around 4.05%) [28] (Figure 10) produced BFO films [28] undergo symmetrical transformation which corresponds with a change of lattice parameters ratio.

There is a definitive change in location of ions in the rhombohedral crystal structure (Fe^+3^ octahedral units) compared to structure with Fe^+5^ cations, suggesting the phase separation into tetragonal and distorted phase of the trigonal symmetry group (R3c). With increasing strain, coefficient films demonstrate metastable R3c–like phase with minimal performance enhancement and phase separation [29]. Summarization of thickness dependence of BFO films on strain coefficient is shown in Figure 10 [28].

It was shown that the remanent polarization of produced films increases with every layer of deposited BFO material, creating a complex multilayer heterostructure (Figure 11). Preferred orientation of deposited multilayers is 110. Improvement of the remanent polarization value is described by the lower parasitic current of the multilayer system, interlayer strain and increased coupling between deposited layers, resulting in the increase of distortion in the BFO perovskite unit.

Typical BFO thin films (Figure 12) can be produced within a wide range of thicknesses. The most common and widely spread range of produced BFO films stays within 50 to 500 nm [28].

### 3.2. Photoelectrical Properties

In addition to a unique combination of ferroelectric and magnetic properties, it is necessary to mention that BFO-based thin films exhibit strong photoelectric properties. Considering the smaller optical bandgap (~2.5–2.7 eV) in comparison to other ferroelectric material (Pb(Zr,Ti)O_3_, LiNbO_3_, BaTiO_3_) and wide range of visible light absorption, BFO material attracts increased attention for potential photovoltaic and/or photogalvanic applications [31]. The photovoltaic phenomenon has been encountered in different forms of BFO material: crystal structure, films and BFO added ceramics. All forms above of the material thin films are extremely useful to be implemented into small- and large-scale devices, since it is uncomplicated to obtain signal from BFO films. Photovoltaic properties of BFO thin films were successfully implemented into various devices, for example, planar photodetectors [32], and complex sensors. Nevertheless, implementation into large-scale devices remains limited considering constrained performance of the material. However, the photovoltaic phenomenon is affected by a variety of factors. A potential attribute of photoelectric properties is photocurrent intensity, which is related to quality of produced BFO films. With smoother and more homogeneous films there is an increase of internal recombination of generated electron–hole pairs. The increase of recombination rate (increase of lifetime) [33] is caused by a decrease of intermolecular (migration) distance within the surface of produced film, which, in its own terms, leads to a potential decrease of current intensity (increase of photoconductivity). Some studies propose that the photovoltaic response is affected by external electric field (polarization) [34]. Direction of photovoltaic response of produced films has been affected by applied electrical field in different directions [34,35]. As shown by Choi and his team, photovoltaic response in single crystal BFO films demonstrate nonlinear behavior and the direction of photovoltaic response can be changed by applied voltage. Furthermore, similar observations of the dependance of photocurrent and electric field has been performed by Yi [36]. In addition to the observation, the theory of the effect of the domain [37] wall angle on photovoltaic response has been proposed. According to the theory, domain walls, which separate holes and electrons, are estimated to be about 100 nm, which is considerably smaller in comparison to conventional silicon semiconductors [38].

As confirmed and observed by Choi [35] and Yang [39] on pure BFO material with periodical domain walls [36], photovoltage largely increases the bandgap of the BFO material [39]. Yang has proven that the strongest photovoltaic response was obtained at the 71° domain wall [34]. Ji and his team have shown that the bulk photovoltaic effect is vital for the determination of photovoltaic response of BFO films (Figure 13) [40]. Furthermore, the photoelectric response is dependent on the structure of produced film. Strain of BFO film could potentially provide enhanced photoconductivity and control over photovoltaic effect of the final film. For example, strained BFO material on LAO [41] exhibits enhanced photoconductivity [42].

Thin films of BFO material show an absorption rate of the wide visible light range of 350 and 575 nm (Figure 14). The wide visible light spectrum means the material can absorb huge amount of light energy [43].

Thin films of BFO material have become more popular for potential photovoltaic applications due to the existence of anomalously large photovoltage, which overcomes the low bandgap of BFO material (~2.7 eV). With the combination of photogalvanic effect, overall chemical stability makes BFO worthy of implementation into photoelectric and photocatalytic devices.

## 4. Deposition Methods of BFO Thin Films

BFO is a unique material which shows ferroelectric and magnetic properties at same time. However, due to the cycloid moment of the cell unit, it is necessary to offset cycloid spin moment to obtain the strongest ferroelectric response. One way to offset cycloid moment is straining material into thin [25] films. It was a crucial point of discussion to find the most cost–effective and suitable method of producing heterostructures based on thin films of BFO material with minimum potential defects.

The most cost–effective method of enhancing overall properties of produced films is the implementation of impurities. It was shown that rare–earth materials dramatically decrease the parasitic current of BFO films because of the compensation of evaporated Bi atoms and oxygen vacancies due to changed overall spin moment of a cell unit. Evaporation of Bi atoms and oxygen vacancies is an unavoidable side effect of production of BFO films. The most promising materials to enhance multiferroic properties and decrease parasitic current are Mn [44] and Ni [45] due to similar atomic radius of Fe cations and chemical valence stability (Table 1).

These dopants aim to substitute cations of Fe^+2^ to compensate residual spin moment, thus lowering the antiferromagnetic response of cell unit. Typical magnetic loop is represented in Figure 15. Recent reports clam the implementation of rare–earth materials lower cell value, thus creating compact surface morphology resulting in a significant decrease in current density.

The typical structure of thin films of BFO material is represented by a multilayer structure (Figure 16). Base substrate layer has a function of carrier of the entire heterostructure of thin film. Most common substrates for the deposition of BFO thin films are conductive and nonconductive glasses (FTO) and a wide variety of ceramic materials. Next, the layer serves as a bottom electrode and buffer barrier of the parasitic current [46]. The crystallographic orientation of the buffer layer will have a determinative effect on the final orientation of produced films. Over the past decade, buffers of Pt/Si/Ti/SiO_2_ materials proved to be great candidates to enhance overall morphology, homogeneity, and lowering leakage current [47].

The most common crystallographic orientation of produced BFO films is a polycrystalline diamond structure (100). In this orientation, deposited films show strong multiferroic properties due to overconcentrated Bi atoms between the bottom electrode and BFO layer. In order to integrate BFO films into potential devices, films are covered by an upper electrode (contacts). Contacts are deposited in the form of big or small dots across surface or as monolayer which covers entire surface of the BFO film. The most prominent material for deposition of contacts is platinum. Pt material has been widely used as a main material for deposition of contacts due to high thermal and chemical stability and superior electrical conductivity.

### 4.1. Sol Gel Deposition Method

Over past decade, the sol gel method has become the most popular method for preparation of thin films in large quantities. The method offers fast deposition rate, low cost of used equipment, and high production rate. Unfortunately, produced films have a significant contamination rate by a high amount of parasitic byproducts, since the sol gel method operates with the presence of oxygen atmosphere and high temperature range. Parasitic byproducts cause a significant increase in the leakage current of produced films and suppression of ferroelectric response. Another drawback of the method is the thickness limitation of final products. Thickness of produced films is varied within several hundreds of nm to µm, which is overly thick to be used in fundamental studies of BFO material. Topography of produced films shows a high rate of pores (Figure 17), which form in places with increased current density resulting in a faster degradation rate of the material.

The sol gel method [24] has been introduced as one of the most spread and earliest methods for synthesizing wide varieties of complex metal oxide particles and thin films. The method is exceptionally good for preparation of films on a large-scale, considering the low costs of equipment and relatively high–quality product outcome.

The method is based on drying of a chemical water–soluble salts precursor with the presence of oxygen atmosphere on the preliminarily cleaned substrate surface. While the precursor is drying down, the composition is becoming denser, turning into a thick gel substance. The method allows precise control over morphology composition, process temperature, and thickness of produced films.

The process starts by preparation of a precursor of water–soluble salts. Most spread compounds of precursor are nitrate and mononitrate salts of bismuth (bismuth nitrate) and iron (iron nitrate) followed by low molecular alcohols, such as methanol, ethanol, etc. These compounds are mixed in highly purified deionized water to lower additional contamination and impurities of resulted films. Once the precursor is prepared it must be rested at room temperature for at least 30 min to settle down. To accelerate the dry down process, additional anhydrites or acids could be added to the final precursor.

Before application (spin coating) (Figure 18) of precursor, the substrates must be properly and repeatedly cleaned by alcohols to obtain a contamination–free surface. The sol gel [48] method allows the utilization of a huge variety of substrates. The most suitable substrates for the sol gel method are conductive and nonconductive inorganic substrates (glass [24], metal plates) due to their chemical and thermal stability against high heat. Organic substrates are suitable as well; however, the morphology of produced films on organic substrates is considerably worse, due to high number of cracks and inhomogeneity of final films.

The application of precursor starts with layering one or several layers of precursor on a cleaned substrate. Each layer of precursor needs to be pre–heated. The preheat is necessary to obtain homogeneous thin films with minimum defects such as cracking and splitting of produced film (Figure 17). Preheating is usually carried out on hot plates with temperatures around 150–235 °C during several minutes. To obtain the required thickness of produced film it is necessary to repeatedly deposit several layers of precursor, preliminarily after pre–heat. The entire duration of complete crystallization of applied layers is around 1 h with temperature around 500 °C and the presence of oxygen atmosphere for maximal crystallization rate.

The versatility of the method allows thin films with different thicknesses to be obtained. The thickness of produced films stays within the range of 100 nm to several μm [17]. With increased thickness of films, overall performance of the final product is rapidly decreasing simultaneously with worsening morphology of resulted films [17]. Since the ferroelectric and magnetic properties of BFO material are size dependent [17], the thickness of produced films is critical for evaluation of performance of final product.

It was shown that the films produced by the sol gel method did not show a ferroelectric response [49], which is given by the decreasing interaction of spin moments of Bi^+2^ and Fe^+3^ ions due to increased intermolecular distance. In addition, a significant decrease of current conductivity was detected. The overall decrease of performance of produced films could be potentially described by the large accumulation of micromorphological structural defects such as cracks, surface decomposition, etc. A huge amount of impurities, which are inevitable considering presence of oxygen atmosphere, have a huge negative impact on electric conductivity.

All that was mentioned indicate the need of optimalization of the sol gel procedure by researching new capabilities to modify process parameters and improve unique properties of resulting BFO thin films [17]. A more suitable method for synthesizing thin films for fundamental research would be either PLD or ALD. Both methods allow producing of films with resulting thickness of several angstroms. The produced films do not undergo extensive contamination by byproducts caused by either the oxygen atmosphere or high heat. These methods, in their principle, use inert gases or vacuum and a low range of operational temperatures to produce homogeneous films with a high degree of stoichiometry. Both techniques may utilize different materials to selectively incorporate various impurities in order to enhance the overall chemical structure of resulted BFO compound.

### 4.2. Atomic Layer Deposition

In the world of continuously growing interest in nanotechnology and multiferroic materials, the discovery of technology which will provide the best final product with outstanding properties and relatively low costs is becoming a hot topic of countless discussions.

Many popular techniques of producing a thin film of Bismuth ferrite either offer low versatility of controlling properties of produced films or operate in a high range of temperatures, which dramatically lower ferroelectric and magnetic properties of Bismuth ferrite films. The technology, which would offer a practical method for producing heterostructures with integrated thin films of Bismuth ferrite and control over every deposited layer, would have enormous potential in near future.

The Atomic Layer Deposition (ALD) [50] method deserves much popularity for preparing complex metal oxides [51] and thin films [52] of Bismuth ferrite for many applications and modern devices. ALD [52] is a chemical method of growing homogeneous thin films at a low range [50] of operation temperatures. ALD [53] belongs to a group of conventional bottom–up type deposition techniques where deposited materials crystalize on surface of a substrate. The method is highly versatile [54], allowing preparation of complicated heterostructures [50] suitable for mostly any potential devices. The ALD method can operate with wide geometric shapes of used samples. In addition to high versatility [53], the method also operates within a relatively low range [52] of temperatures (200–400 °C) which is a considerable advantage for producing films [55] of BFO material. The low range of temperature enhances magnetic and ferroelectric properties of a final product due to the low rate of evaporation of Bismuth atoms.

ALD [53] is based on a self–limited repeatable reaction between a gas precursor and substrate surface. By stocking every deposited layer, a heterostructure will be the final product of the synthesis. The deposition is carried out inside a chamber with the presence of a gas precursor under vacuum (10^−6^–10^−9^ Torr). Considering the versatility of the ALD [55], the method allows for single layer deposition of a certain material per single cycle [52]. The duration of a single cycle stays around 30–60 min. To prevent potential contamination of monolayers and corrosion by an aggressive chemical precursor, the main chamber is ventilated by an inert gas (Ar or N_2_) [48]. The cycle is repeated over again in order to obtain the required thickness of the resulted heterostructure. Figure 19 illustrates the process of the ALD method. The process is carried out at lower temperatures (max 400 °C) to prevent potential oxidation of monolayers [55]. The lower the temperatures, the less potential surface defects such as surface decomposition, oxidation, and uneven surface morphology will occur [48]. Considering the high evaporation rate of bismuth atoms during preparation process, a low operational range of temperatures [56] is vital to achieve better quality produced thin films [54] of BFO [57] material.

The fundamental advantage of produced monolayers by the ALD process is the homogeneity of single layers over a large surface area, [54] which is a crucial advantage of the method making ALD [52] an outstanding method of producing heterostructures [48]. The homogeneity is achieved by a gas precursor, which will spread evenly throughout the surface and deep trenches, and thus fully reacting with entire surface of the substrate [48].

The next considerable advantage is a wide thickness [58] range of the resulted films. The ALD process offers a wide range of thicknesses of produced films. The thickness range [58] is determined by the number of repeated deposition cycles [48]. The growth rate of the single cycle stays within several angstroms to hundreds of nanometers and is based on the duration of deposition cycle and individual requirements [48]. The thickness versatility of ALD is a considerable advantage for the preparation of thin films for numerous applications.

### 4.3. Pulsed Laser Deposition

In the world of nanotechnologies, the trend of producing high–quality nanomaterials and small dimensional lithographic structures is of current interest. The conventional methods of preparing thin films are divided into two different types: top–bottom and bottom–up [19]. Bottom–up methods offer a wide range of possibilities of producing a huge variety of nanostructures. These methods also influence morphology, orientation, and properties of produced nanostructures by changing operational parameters.

Laser technologies have been successfully used in many different applications over many years. The laser has a unique combination of outstanding properties such as monochromatic, pulsed and continuous operation modes, and a narrow energy distribution making it superior for a wide variety of applications.

Pulsed Laser Deposition (PLD) [59,60] earned a title to be an extremely good and versatile method for the deposition of a wide range of complex heterostructures and high–quality thin films of many materials due to the selective evaporation of practically any materials. The first films were synthesized in 1965 by scientists Turner and Smith [61,62]. The laser with ruby medium was used as a primary instrument to synthesize the first thin films. The year 1965 is a starting point of the development of the PLD method. After successful deposition of thin films, the PLD method started gaining deserved attention and popularity.

The principle of PLD is based on the interaction of laser radiation [61,62] with the liquid of solid material (target) [59,60], which leads to the absorption of laser radiation. After irradiation of target with a laser beam, a small amount of target material is evaporated and carried away from the target. The result of the absorption is the ablation of particles which are carried away with kinetic energy.

The evaporation rate of microscopic particles and kinetic energy of evaporated particles are proportional to the laser operational parameters. Depending on the kinetic energy [63,64] of evaporated material, the increase of interaction will require a cooler temperature range to compensate for the collusion of evaporated particles, which would lead to an increase of the deposition rate. The main parameters, which effect the evaporation rate [61] of the targeted material, are energy intensity, pulse duration (period), laser wavelength and angle of incidence between laser beam [65] and target material. The evaporated material comes into contact with cooler substrate, where films will be deposited, and condensates on substate surface, creating the required films. The laser system repeats pulsation, keeping the evaporation rate stable and synthesizing the desired thickness of resulted thin films. The main chamber and laser system are represented in Figure 20.

The process starts by interaction of a powerful laser [59] with the targeted material of the BFO compound. The evaporated component will condensate on the surface of a substrate. Popular substrates, which are used for the preparation of BFO thin films [57], are polycrystals of Si with different orientation {111, 110, 001} (Figure 21), Dysprosium Scandium Oxide (DyScO_3_), and Strontium titanate (SrTiO_3_, STO substrate). Typical operational parameters of the PLD method include repetition rate, deposition time, substrate preheat temperature, laser pulses and main chamber pressure. The orientation of substrate will be a decisive factor of the resulted BFO crystal orientation. Since the substrate is targeted by a laser, the crystallization of BFO is started by creating a crystal matrix of Bi_2_O_3_ and Fe_2_O_3_ materials with a different orientation. The pulse rate repetition stays around 5–15 Hz [61,62] and may vary based on the recipe. The wavelength of laser radiation usually corresponds to a deep ultraviolet color (200–400 nm), which is the most used laser radiation. To achieve maximum energy distribution on the targeted material the incidence angle should stay around 45° [66]. The entire process is accompanied by vacuum and/or the presence of an inert gas to reduce potential impurities. The duration of the deposition process is based on the thickness of produced film and pulses of the laser. The typical duration of the deposition cycle of 100nm film with 5000 pulses stays within 20 min. The deposition temperature of BFO films (substrate) stays under 500 °C.

PLD has considerable advantages compared to the conventional techniques of the preparation of thin film materials based on BFO material.

The major superiority of PLD [64] is stoichiometry of produced films. Stoichiometry is caused by a high condensation rate of evaporated particles of targeted material on the substrate surface. The versatility of the operation process allows changing targeted material sequentially on a rotatable holder which is a key advantage of the process. Sequential change of targeted material is necessary to produce multi–layer complex structures and the development of new artificial structures of stable/metastable materials without inflicting aggressive operation conditions by changing laser parameters. An example of the multi–layer structure of BFO material is represented in Figure 22a,b.

The operational parameters of PLD, such as laser optical energy [64,67] distribution, the distance between target and substrate, the substrate temperature and grow rate greatly constrain the negative effect of morphology and enhance magnetic and electrical properties of the resulted thin film structures.

In comparison to costly, conventional methods (selective ion implementation) which operate in a higher temperature range, thus producing various types of defects and impurities of deposited composite, the PLD [64,67] method operates in a lower temperature range.

The price of used substrate is another disadvantage of conventional methods. Considering that the PLD can operate in a lower range of temperatures, the PLD method operates with more available and cheaper polymer base substrates [67].

It has been reported that films produced by the PLD method show relatively low parasitic current, considering the low amount of crystallographic defects and small particles size [68].

### 4.4. Comparison of ALD, PLD and Sol Gel Methods

By comparing the most widespread methods of the synthesis BFO thin films, the sol gel method was proven to be the most popular method for the preparation of films in huge quantities. In comparison to PLD and ALD, the sol gel method (Table 2) is relatively inexpensive since it does not utilize a vacuum nor inert gases during the preparation procedure. The major downside of the sol gel method is a significantly high defect rate of the crystallographic and morphological structure of produced films. The quality of produced films is based on purity of used chemical ingredients such as water–soluble salts and chemical solvents. The presence of an oxygen atmosphere has a negative effect on the properties of BFO films due to occurring oxide side products. Thin films produced by the sol gel method exhibit a high parasitic current, which is given by the contaminated surface due to the presence of oxygen atmosphere and impurities of chemical compounds. In addition, films demonstrate a negligible ferroelectric response given by increased particle size. The sol gel method would be extremely useful for the preparation of films for applications where quality of morphology and properties are tolerable such as range scale sensor devices. The sol gel method does not show versatility of producing complex heterostructures of BFO material, and instead allows for the preparation of “sandwich” structures with different monolayers.

The operation principles of PLD and ALD methods share many similarities. Both methods use an external heat source to evaporate a small amount of targeted material. The evaporated material will condensate on a cooler substrate surface, creating the demanded orientation of thin film. Both methods show a high degree of quality of produced films by operating with the presence of a vacuum and inert gases to further decrease the contamination rate and impurities of produced films. The methods do not involve the operation with presence of aggressive chemical solvents, which could have a negative effect on produced films. One major advantage of both methods is their operation within a low temperature range (<350–400 °C), in comparison to the sol gel method (400–700 °C). The produced thin films by PLD and ALD methods show considerably lower parasitic current and a high degree of stoichiometry. The key advantage of both methods is the effective deposition of complex heterostructures with different crystallographic orientation with different material monolayers. Both methods offer a wide range of thicknesses of produced films. The thickness of the final product varies from several angstroms to hundreds of nanometers where minimal thickness of produced films by the sol gel method is 100 nm and maximum may increase several µm. The versatility and high quality of produced films comes with increased coasts of both methods, which is a significant disadvantage of these methods. The high cost of both methods is given by the utilization of laser technologies, vacuum and inert gases, expensive materials, and equipment. Both methods are suitable the preparation of thin films in small quantities with superior properties and morphology for precise measurement devices or prototypes.

## 5. Side Phases of Bismuth Ferrite

The need for cost–effective energy sources involve seeking new and prospective materials and technologies. Considering ever increasing pollution and rapid development of the manufacturing industry, the use of solar energy has become an undivided part of the modern trend of utilization of eco–friendly energy sources. Nowadays trends of designing microelectronic devises offer new challenges, involving the design of microelectronic devices which would be based on manipulation by electric and magnetic parts. These challenges require investigation and recache of new materials. Among all materials, multiferroics and especially Bismuth ferrite, stands out. The BFO material aims to be a potential solution regarding simultaneous control of magnetic and electric parts, since BFO exhibits the strongest ferroelectric and magnetic response at room temperature.

### 5.1. Bismuth Mullite Material

During intensive studying of Bismuth ferrite material preparation methods, it was found that during the initial preparation process of BFO powders, alongside pure BFO material, other side–phase materials were synthesized. It was shown, due to the aggressive chemical nature [4] of bismuth, that BFO causes byproducts. These materials have been categorized as impurities of the BFO material. The two most popular byproducts of synthesis of pure BFO material are bismuth Mullite and Iron Selenite. Instead of crystalizing the single–phase of the BFO material, multi–phase side products were crystalized [69]. The mechanism of crystallization side products is related to the stoichiometric imbalance [70] of BFO [71]. The stoichiometric imbalance refers to the asymmetrical growth of FeO monomers (some particles could evaporate during crystallization), which are particles that can undergo side reactions. The chemical equation Bi_x_Fe_y_O1.5_x_ + 1.5_y_ [72] describes the general composition of a single–phase BFO [1] material and secondary phase of new side products.

The first and most prominent byproduct [73] of the synthesis pure BFO material is mullite ferrite Bi_2_Fe_4_O_9_. The typical cell unit of the mullite material is represented in Figure 23. The crystallographic structure of the mullite is represented by a central symmetric [74], orthorhombic structure [73] with two atoms of Bi^+2^, four atoms of Fe^+2^, and 9 atoms of oxygen [75] with different valency. The lattice parameters are a=b=8 Å [1], c=6 Å [74] and the cell value is around 400–410 Å [74,76]. The parameters may differentiate due to structure defects during the preparation process, and oxygen vacancies [74]. A typical Bi_2_Fe_4_O_9_ [77] cell unit consists of FeO_4_ tetrahedra [73,75] and FeO_6_ octahedra [74] formation. The octahedral [78] bond distance is approximately 2 Å, and the tetrahedra distance is smaller and stays within 1.8 Å [74,79] (Figure 24).

Fe^+3^ cations are in both tetrahedra and octahedra formations. FeO_n_ are distributed homogeneously across the unit cell and are surrounded by Bi ions. Spin moments of octahedral [81] FeO_6_ create couples with electrons of tetrahedral the FeO_4_ spin moments. The coupling of FeO_n_ spin moments lead to a stronger interaction between the two monomers, thus causing strong anti–ferromagnetic properties [82] of the unit cell (G–type ordering). The anti-ferromagnetic response is given by the interaction of coupled electron spin moments of FeO_4_ and FeO_6_ monomers and uneven spin distribution, causing overall geometrical distortion of spin moments [82].

It was reported that the mullite also exhibits anti–ferromagnetic properties closer to the 0 °C [73] temperature. The paramagnetic transition temperature to antiferromagnetic (Neel) of the compound stays within −23 °C as Curie temperature was determined to be in the range of −9 to 0 °C. In addition to anti–ferromagnetic properties, the mullite ferrite demonstrates ferromagnetic ordering. Ferromagnetic properties are observed by parallel orientation of spin moments across one dimensional axis. Ferroelectric properties of mullite ferrite are potentially described by asymmetrical distribution of hybridization between s– and p– orbitals of bismuth cations and oxygen vacancies.

However, overall asymmetrical frustration might be stabilized by the interaction of SP hybridization of Bismuth ions and oxygen anions.

Magnetic and ferroelectric properties (response) (Figure 25) of the mullite material are size dependent and show similar behavior to pure BFO material (Figure 26). It was shown that the smaller particles demonstrate a stronger ferromagnetic response, in comparison to greater particle size. The increase in ferromagnetic properties could be described by a decreased interaction of tetrahedral and octahedral spin moments, changing the spin layout of a cell unit.

Recent reports claim the ferroelectric response is negligible, considering the symmetrical orientation of unit cell by the central axis. Therefore, it is necessary to find a suitable solution to enhance the magnetic and ferroelectric properties of the mullite for future implementation into multifunctioning devices. A potential solution would be adding impurities of rare–earth material to compensate oxygen vacancies. It was proven that rare–earth material, such as lanthanum and magnesium, dramatically enhance ferroelectric properties [73].

Mullite is widely spread [83] in chemistry and sensor devices. The material is extremely sensitive on alkali vapors and was successfully used as a gas [84] leakage sensor. The mullite proves to be a great and cheap catalyzer for chemical reactions, such as the decomposition of ammonia to nitrate oxide.

Recent research reports the existence of different reactions of visible–light absorption, which suggests the material possesses photocatalyst [85] and photoelectric properties. Hence, Mullite can be used for the utilization of solar [86] radiation. The material shows wavelengths absorption within the range of 350–700 nm [86] (Figure 27). The range of wavelengths corresponds with 2–3 eV [79].

The photocatalytic [85,87] properties prove that the material has a lower bandgap potential [88], which is a considerable advantage, and make the material attractive for light detection sensors and photovoltaic application [88].

Mullite is a proven, very promising material to be implemented into new types of sensors. Mullite is widely used in the area of sensors for detecting gas leakage due to its chemical nature, and in organic chemistry as an enhancer of chemical reactions [1]. The full potential of the material is still not revealed; thus, more and deeper studying is required.

### 5.2. Iron Selenite Material

Bismuth ferrite is represented in two different structures: perovskite cell and Selenite material. Iron Selenite material (Bi_25_FeO_40_) [87,89] has been encountered during the synthesis of pure phase BFO material. Selenite draws a lot of attention nowadays due to its unique and strong photocatalytic [90] properties [91], which is a popular subject of research. Selenite belongs to semiconductor family and was proven to be a very eco–friendly material for a wide variety of applications. In addition, Selenite has become an outstanding candidate for solar light utilization technologies due to the slower rate of sunlight degradation, low bandgap potential [91,92] and overall harsh chemical environment resistance. A superior paramagnetic response is another advantage of the material, making it easily obtained and separated from pure BFO material during the synthesis process. The existence of a hysteresis curve near room temperature range suggests ferroelectric and dielectric properties of Selenite material.

Selenite material (Figure 28) occurs as another side product of the synthesis of pure BFO material. It was proven by many experiments [89,92,93] that the most widespread methods to obtain pure phase Selenite material are the sol gel and hydrothermal grow methods. Selenite is usually obtained from a chemical precursor of bismuth nitride and iron nitride water–soluble salts with an operational temperature around 650–750 °C [92,94].

The crystallographic structure of the Selenite unit (Figure 28) is represented by cubic asymmetrical I23 space orientation with lattice parameters a = b = c = 10 Å and the estimated crystal value is about 250 Å. In this orientation bismuth cations with valence Bi^+3^ [92], together with surrounding oxygen atoms, form incomplete BiO_5_ octahedral units. The interatomic distance between Bi^+3^ and shared oxygen atoms is around 2–2.5 Å. The octahedral unit is completed by the inert 6s^2^ [92] pair. An anisotropic vibration is shown in Bi^+3^ atoms. Each Fe^+3^ [92] and Bi^+5^ [92] cation forms tetrahedral [92] units. Bi^+5^ ions cause distribution not only of the tetrahedral unit, but also of the entire chemical structure. Bi^+5^ [92] cations are present in the cell unit which leads to the chemical formula of Bi_24_^+3^(Bi^+5^Fe^+3^)O_40_ [92] with stoichiometric formation.

Selenite shows weak ferromagnetic activity with transitional temperature, around –5–0 C° [92], and significant spontaneous magnetization at room temperature.

Selenite is considered a perfect candidate for the photo–Fenton [95,96] reaction enhancer. The photo–Fenton [95,96] reactions stand for advance oxidation where organic compounds are decomposed and disinfect in water. Nevertheless, insufficient surface transition of Fe^+2^/Fe^+3^ [92] limits the photo–Fenton [95] (Figure 29) reaction enhancing activity of the material. Iron Selenite shows remarkable properties, which are useful in piezoelectric [92] and especially in photo [97] applications.

Considering even the lower band gap [99] of Iron-Selenite, in comparison to pure Bismuth ferrite (2–3 eV [100]), the material was proven to be an exceptionally good absorbent of ultraviolet [101] and visible light radiation attracting much attention. The band gap [102] of Iron Selenite stands under 2 eV [92], which is much lower among most of the conventional photo–active materials. By analyzing the absorption [99] activity, it was found that the absorption rate of Iron Selenite is even higher compared to Mullite Ferrite. The wavelength absorption [103] range of Mullite stays within 600–850 nm, whereas for Iron Selenite it is much wider within 500–900 nm [99,101] (Figure 30), and with a smother absorption rate at higher wavelengths, resulting in stronger optical absorption capabilities.

In recent years, Iron Selenite (Figure 28) was widely used in digital logic and sensor devices due to the existence of a hysteresis loop. A hysteresis loop suggests the existence of magnetic and electric coupling which could be changed by applying a magnetic and/or electric field. The paramagnetic behavior of the material is another aspect which makes the material a potential candidate for digital applications.

The perovskite–like crystallographic orientation of Iron Selenite and chemical structure of the material open new opportunities for the implementation of a wide variety of impurities. Impurities are necessary to compensate for vacancies of evaporated oxygen and/or Bismuth atoms, which is major issue of conventional, high–temperature deposition methods. Impurity changes the overall spin structure, thus enhancing the unique ferromagnetic and ferroelectric properties of the compound.

Although the Selenite material was widely studied, the full potential of the material is not yet revealed and requires further and deeper investigation.

## 6. Discussion

BFO material is an outstanding material, which combines a variety of unique properties, including high remnant polarization, narrow bandgap potential ferroelectric, magnetic coupling, etc. In addition to unique ferromagnetic properties, Bismuth ferrite shows this unique combination at room temperature. This material offers versatility of exploration of new technologies with complex functions. Despite having a unique combination of ferroelectric and magnetic properties, the high parasitic current is significant downside of the material, which limits its implementation into large-scale devices. Many strategies were offered to enhance and modify overall performance and decrease parasitic current. New low–temperature deposition methods (PLD, ALD, etc.), proper implementation of rare–earth materials impurities and straining BFO material into thin films achieve great results of enhancing performance of BFO material. The implementation of new strategies drives the utilization of BFO material in different fields, such as photovoltaic, piezoelectric, sensorics and memory devices. Nevertheless, even though great results have been achieved, many challenges remain. It is necessary to further enhance the magnetic response of the material before incorporating it into large-scale devices. The crystallographic structure of Bismuth ferrite is represented by a rhombohedral perovskite structure. In this structure, the material exhibits a ferroelectric response due to the residual magnetic spin orientation of a single cell unit. The perovskite structure offers opportunities for implementation of different materials to improve an overall spin structure by compensation of decreased valency of Fe atoms, resulting in stronger ferroelectric response and decreased parasitic current. The parasitic current is given by defects of crystallographic structure of Bismuth ferrite during the initial deposition method. One of the most common causes of parasitic current is evaporation of Bi atoms and a decrease of valency of Fe atoms. A high operational temperature resulting in a high evaporation rate of chemically volatile Bismuth, suggests that the modification (or choice) deposition method operates with the lower temperature range. The best suitable deposition methods for preparation of Bismuth ferrite material are Pulsed laser Deposition (PLD) and Atomic Layer Deposition (ALD), since they operate in a relatively low temperature range and offer outstanding control over morphology of final products. A low range of operational temperatures is extremely important for the evaluation of ferroelectric properties of the produced BFO material, considering the chemical volatility of Bismuth atoms. Considering the perovskite–like structure, the impurities of different material aim to enhance electromagnetic coupling of BFO. Impurities of rare–earth materials not only stabilize the crystal structure of BFO but also dramatically increase the magnetic response due to closer interaction of spin moments of the entire cell unit. Implementation of rare–earth materials considerably increase the transition between ferromagnetic and antiferromagnetic phases due to the elimination of the cycloid spin moment, allowing a strong magnetic response to occur. Most studied impurities of rare–earth materials are Titanium, Neodymium, Lanthanum, Samarium, Europium, Praseodymium; implementation of these materials causes a significant increase of electromagnetic coupling [105,106]. Another potential solution to enhance ferroelectric response is a decrease in cell unit size. The decrease of cell size could be achieved by straining the BFO material into thin films by applying pressure, which leads to a transformation between ferromagnetic and antiferromagnetic phases (morphotropic phase boundary). Smaller cell units show greater ferroelectric response due to a stronger interaction of spin moments of the single cell unit. The existence of the morphotropic phase transformation under applied mechanical pressure allows the discovery of new ways of enhancement of overall ferroelectric properties, which could dramatically decrease production costs of new types of ferroelectric materials for potential complex devices. The most versatile form of BFO material, which offers a wide variety of produced particles and modifications, is thin film. Among other forms of BFO material (powders, nanoparticles), thin films are exceptionally versatile and can be easily integrated into large-scale devices. In addition to versatility, different materials could be implemented into the thin film structure to enhance the unique properties of BFO material. Besides thin films, BFO material was extensively used for the preparation of BFO-based ceramics due to its temperature stability. BFO is a lead–free, non–toxic material with the electrical structure of bismuth molecules being similar to Pb. The perovskite–like structure of BFO allows for the creation of the morphotropic phase transformation enabling the achievement of maximized piezoelectric response. Nevertheless, problems with the preparation of BFO–based ceramics associated with high operation temperatures and unfortunate combination of low electrical resistance and coercive field of BFO material cause difficulties with research of the piezoelectric response and domain behavior. These reasons were at the beginning of numerous research avenues of potential modification of BFO–based ceramics. The most interesting compounds are BaTiO_3_, Bi–K–TiO, Bi–Na–Ti–O, and Bi(Zn–Ti)O, since these compounds enhance Curie temperature and piezoelectric response. Unfortunately, the properties of these compounds have not been systemized.

Byproducts of the BFO material crystallization were discovered during the initial deposition process. Mullite and Iron Selenite are two common byproducts, which were under intense investigation, considering the uniqueness of their properties. The strong photoelectric response and visible light absorptions capabilities of both byproducts was proven vital for photovoltaic applications and devices.

## 7. Conclusions

This paper provided a review of the BFO material. The reasons for making Bismuth ferrite unique over other multiferroic material has been described. The crystallographic structure of the material is represented by a cubic Rhombohedral perovskite structure. This paper describes the crystallographic structure of BFO material in detail, including the spin structure of a single cell unite and their effect on ferroelectric and magnetic properties. The main disadvantage of the material is the parasitic current. The reason for the occurrence of parasitic current and potential solutions of how to decrease parasitic current have been described. Bismuth ferrite is synthesized in different forms, such as powders, nanoparticles and thin films. Thin films of the BFO material are the most widespread and popular form of the material. Morphology, properties, variety of thicknesses and their effect on leakage current have been described in detail.

The deposition methods of thin films include: the sol gel method, Pulsed Lased Deposition (PLD), and Atomic Layer Deposition (ALD). Their operational principle, comparison, advantages, and disadvantages for preparation of BFO thin films have been mentioned. Among the described methods, ALD and PLD are superior methods, allowing the deposition of a single atomic layer stoichiometric heterostructure of thin films with minimal contamination.

It was discovered that thin films include byproducts of Bismuth ferrite. Iron Selenite and Bismuth Mullite are the typical byproducts of the initial synthesis of BFO films. The chemical and crystallographic structure of both byproduct materials have been described. Superior visible light absorption, and strong photoelectric properties have been mentioned.

Future perspectives of the thesis will lie with analyzing samples of the BFO material. Samples will be prepared by the PLD method on ceramic substrates with a platinum buffer layer. Analyzing methods include topography analysis by atomic force microscopy (AFM) and near–field scanning optical microscopy (SNOM), due to their availability at the research facility and uncomplicated scanning procedure. In addition to AFM and SNOM microscopy, topography of prepared samples will by studied by scanning electron microscopy (SEM) and focused ion beam (FIB); both methods will provide detailed information of cross section and thickness of BFO and platinum buffer layer of produced samples. To determine whether there are any byproducts present in produced samples, Raman spectroscopy will be extensively used. In the case of discovery of potential byproducts of the BFO material the attempt of extraction will be carried out.

## Figures and Tables

**Figure 1 materials-15-08719-f001:**
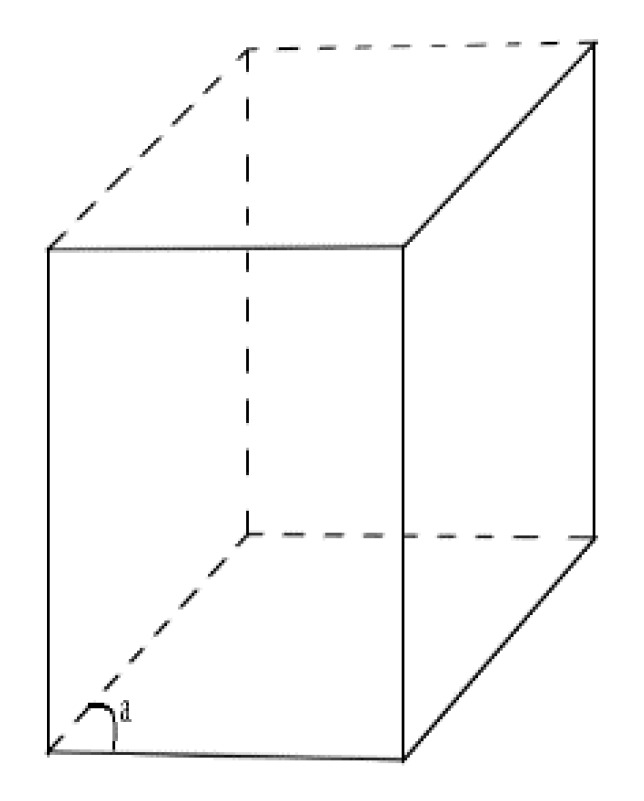
Rhombohedral crystallographic structure.

**Figure 2 materials-15-08719-f002:**
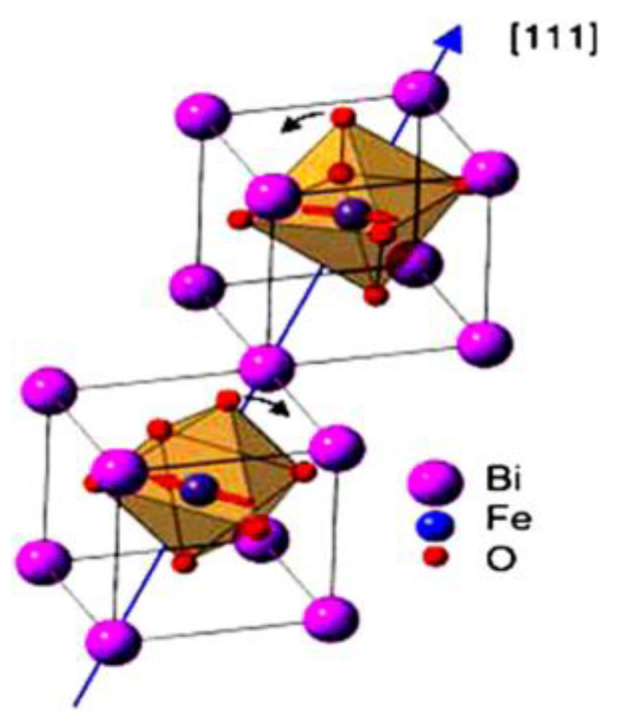
Bismuth ferrite unit in the perovskite structure [8].

**Figure 3 materials-15-08719-f003:**
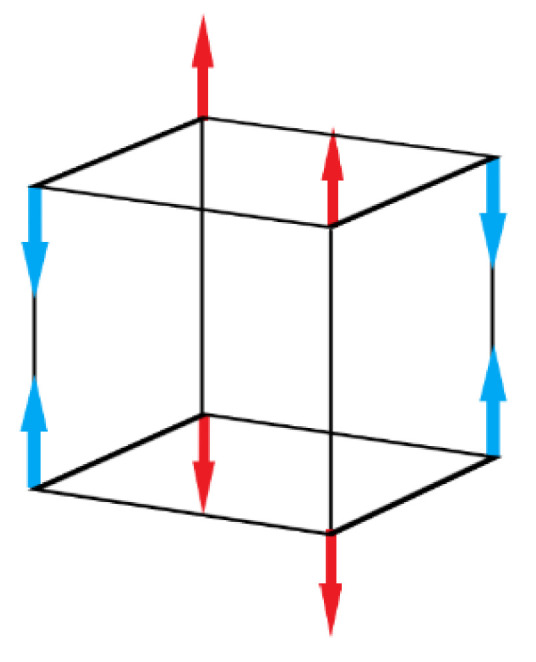
G–type magnetization order.

**Figure 4 materials-15-08719-f004:**
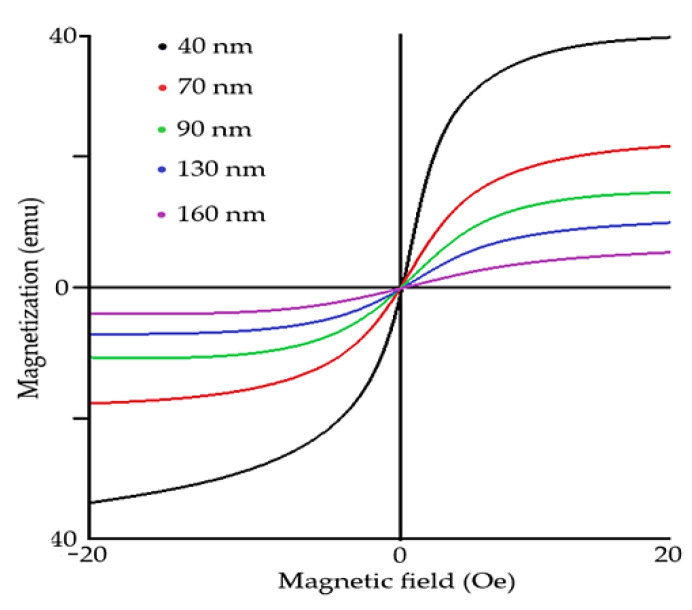
Hysteresis loop of various particle sizes of BFO material.

**Figure 5 materials-15-08719-f005:**
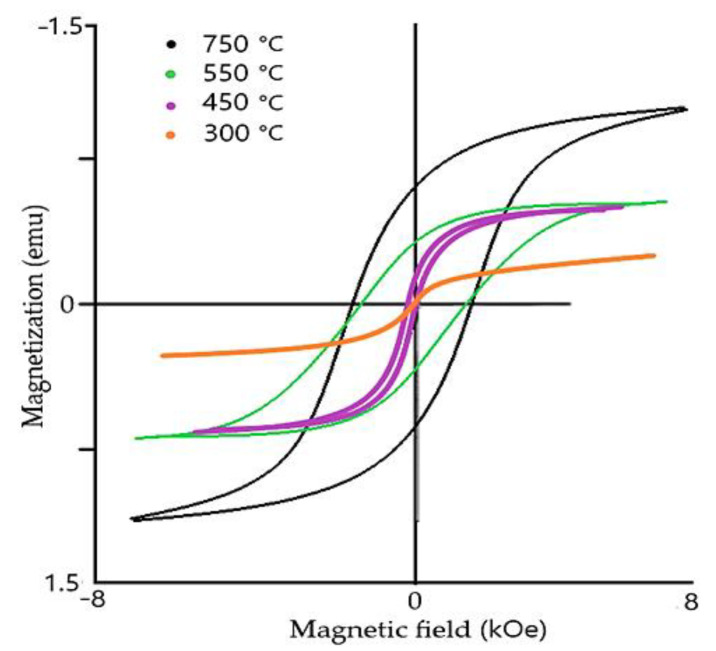
Temperature-dependent magnetic hysteresis loop of BFO material.

**Figure 6 materials-15-08719-f006:**
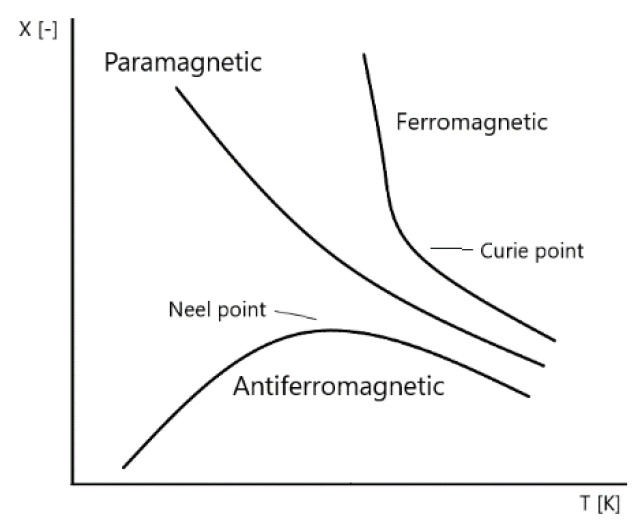
Magnetic susceptibility vs. temperature curve.

**Figure 7 materials-15-08719-f007:**
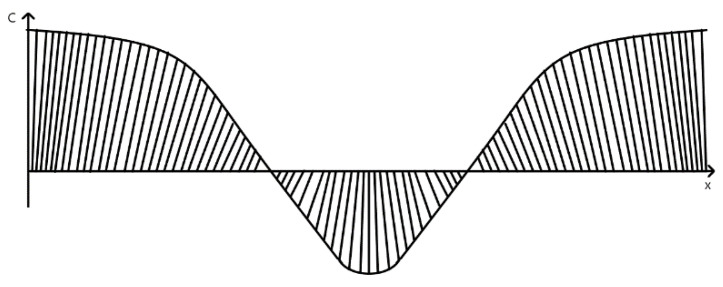
Antiferromagnetic curve of BFO.

**Figure 8 materials-15-08719-f008:**
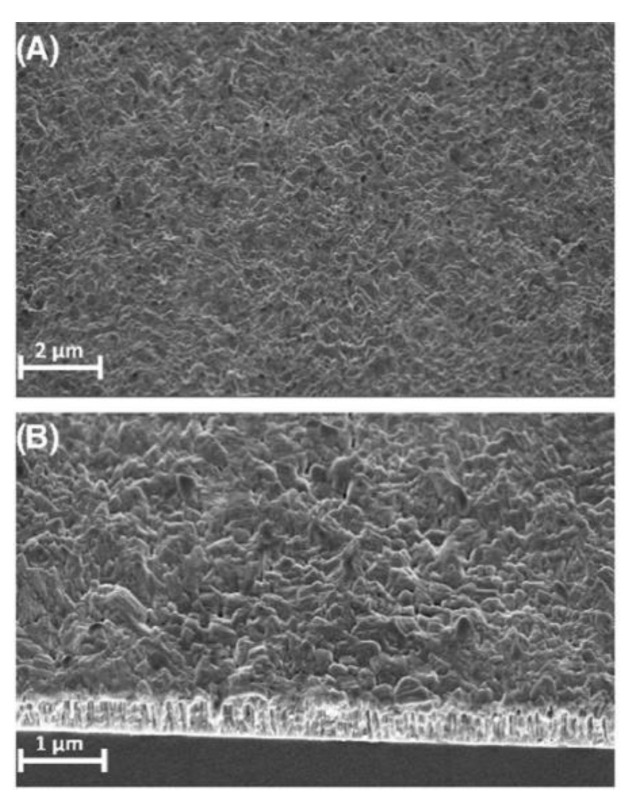
Morphology of a BFO thin film: (**A**) 2 µm; (**B**) 1 µm [25].

**Figure 9 materials-15-08719-f009:**
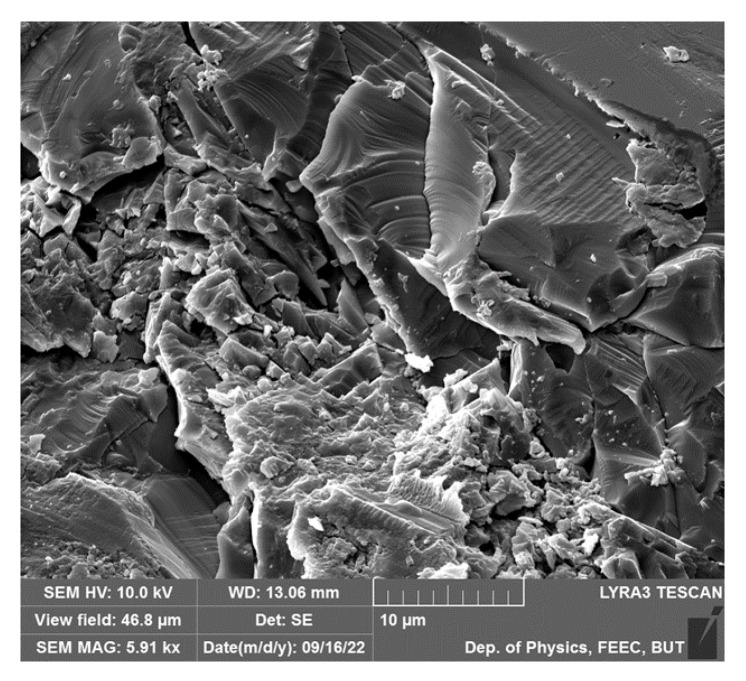
BFO sample surface.

**Figure 10 materials-15-08719-f010:**
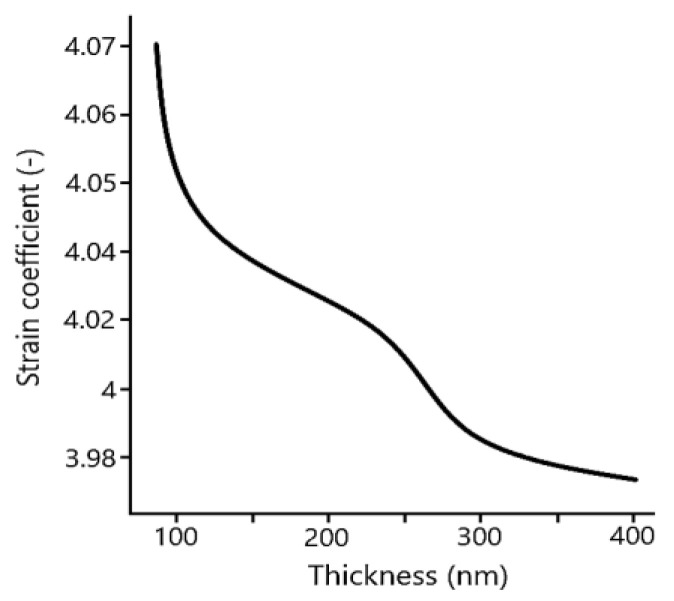
Strain coefficient dependence on thickness [28].

**Figure 11 materials-15-08719-f011:**
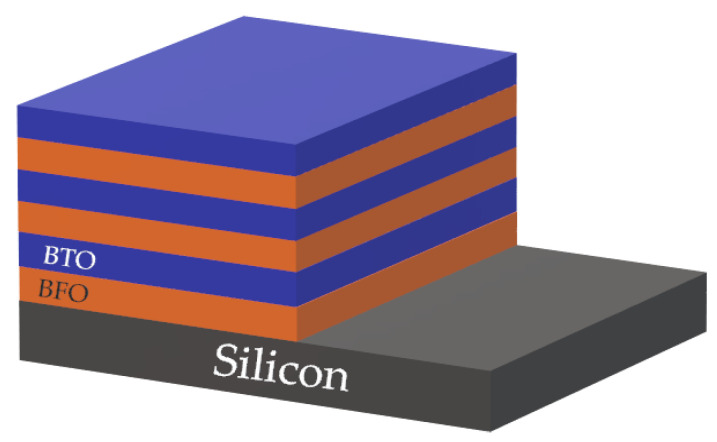
Multi–layer heterostructure of BFO BTO materials.

**Figure 12 materials-15-08719-f012:**
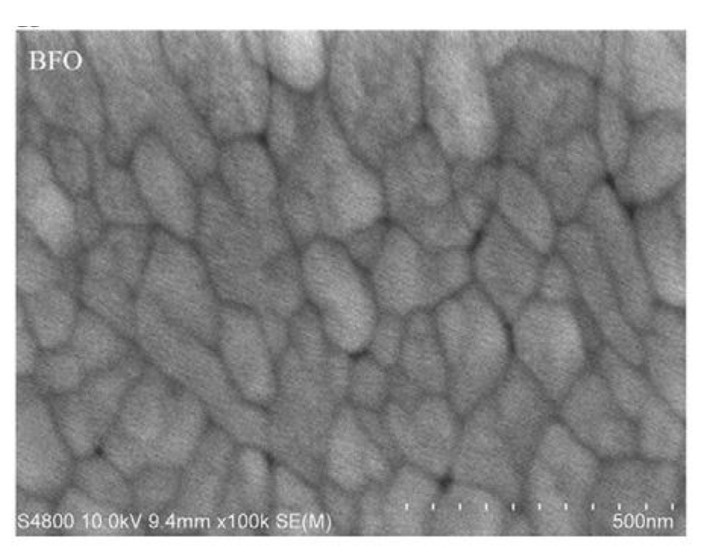
Pure BFO film surface morphology [30].

**Figure 13 materials-15-08719-f013:**
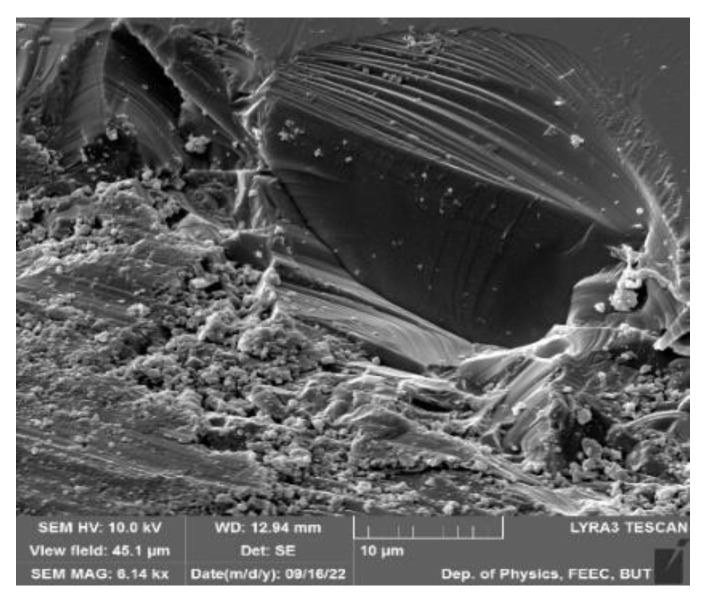
Sample of deposited BFO material.

**Figure 14 materials-15-08719-f014:**
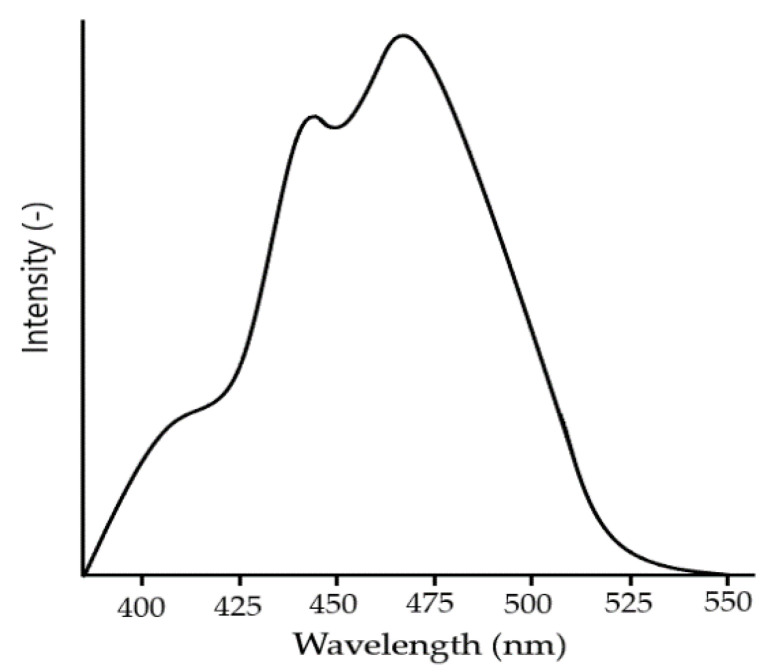
Absorption rate of visible light wavelength of BFO material.

**Figure 15 materials-15-08719-f015:**
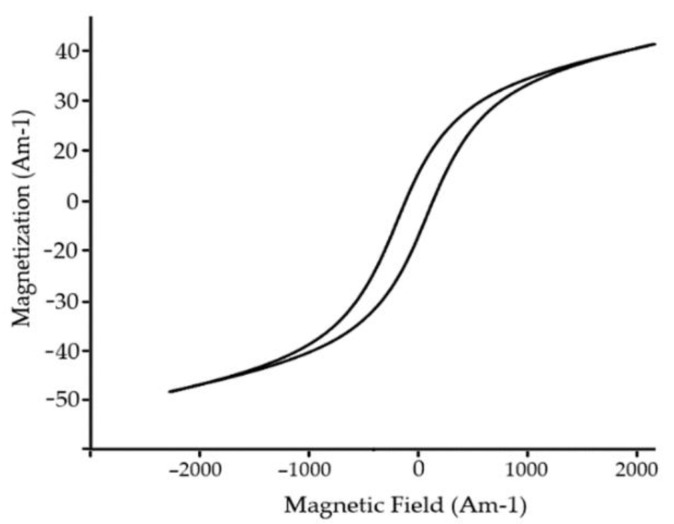
Typical magnetic loop of BFO thin film.

**Figure 16 materials-15-08719-f016:**
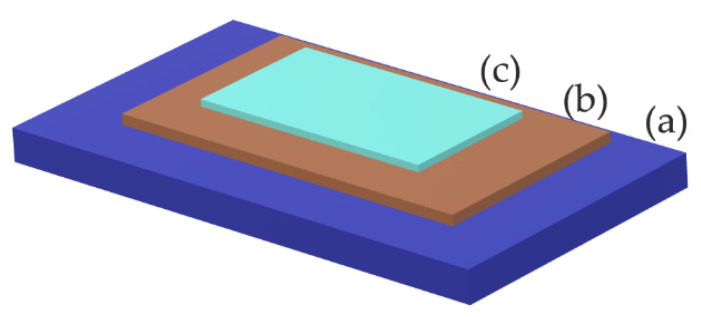
Structure of a BFO thin film (**a**) substrate; (**b**) buffer layer; (**c**) Bismuth ferrite.

**Figure 17 materials-15-08719-f017:**
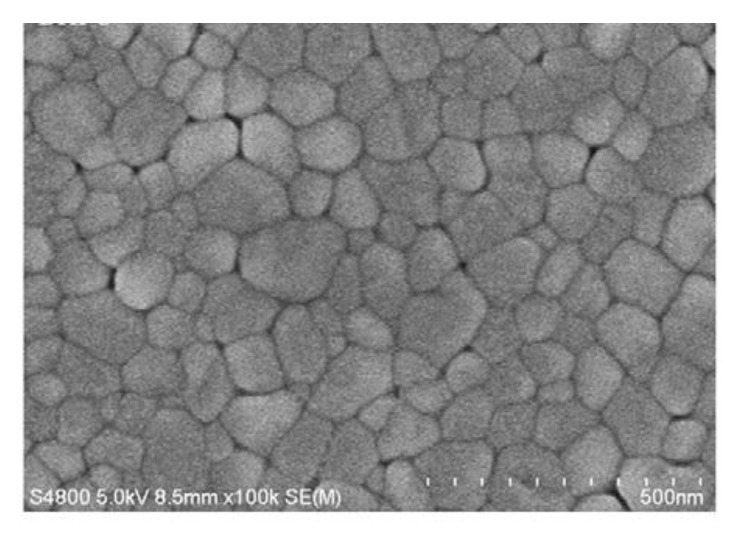
Topography of BFO film produced by the sol gel method [30].

**Figure 18 materials-15-08719-f018:**
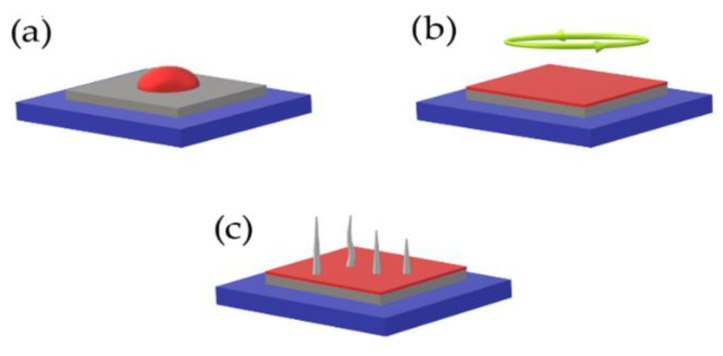
Sol gel process representation: (**a**) deposition of precursor material; (**b**) spin coating; (**c**) preheat and dry down.

**Figure 19 materials-15-08719-f019:**
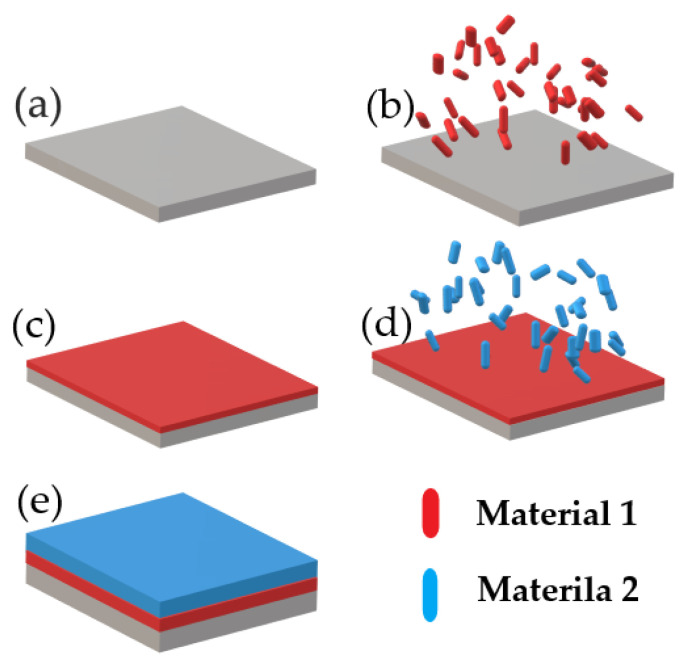
ALD Process diagram: (**a**) prepared substrate; (**b**) evaporated particles of Material 1; (**c**) deposited Material 1; (**d**) evaporated particles of Material 2; (**e**) deposited Material 1.

**Figure 20 materials-15-08719-f020:**
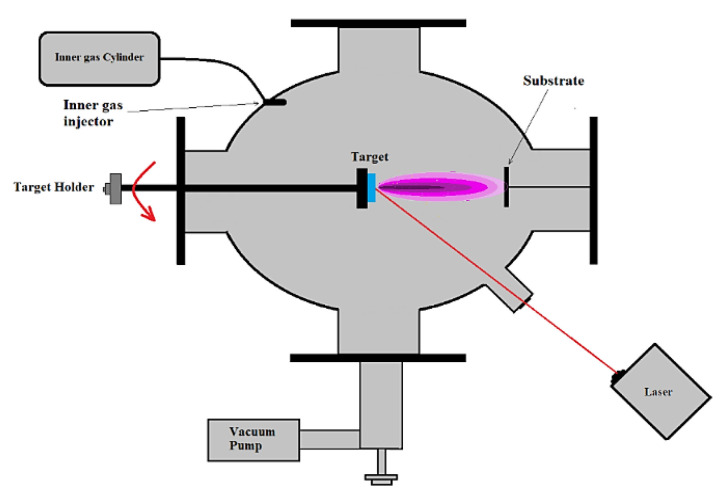
PLD laser construction setup.

**Figure 21 materials-15-08719-f021:**
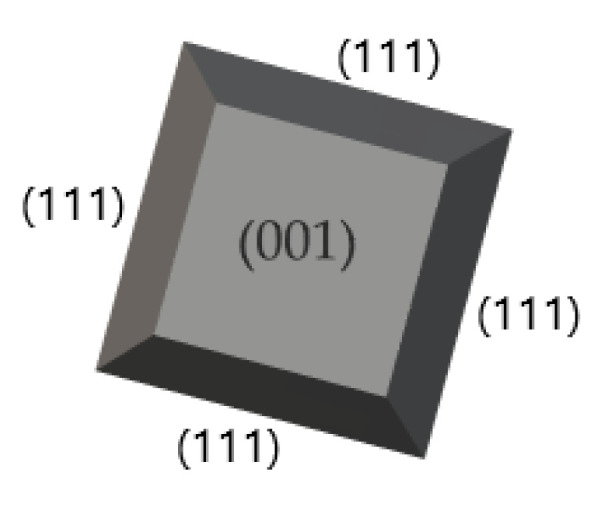
Crystallography of produced crystals based on the orientation of substrate.

**Figure 22 materials-15-08719-f022:**
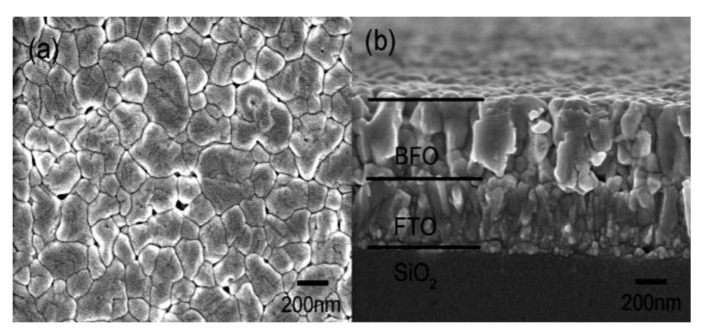
SEM image of a BFO film: (**a**) topography; (**b**) cross–section [67].

**Figure 23 materials-15-08719-f023:**
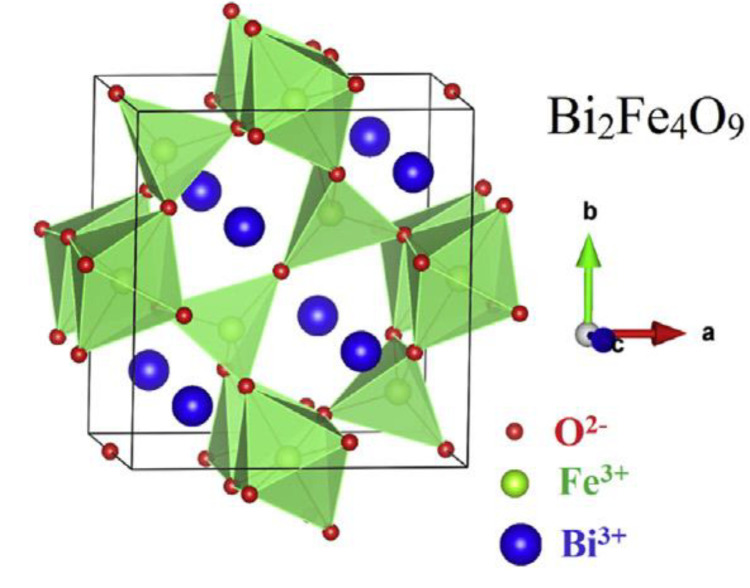
Cell unit of Bi_2_Fe_4_O_9_ [80].

**Figure 24 materials-15-08719-f024:**
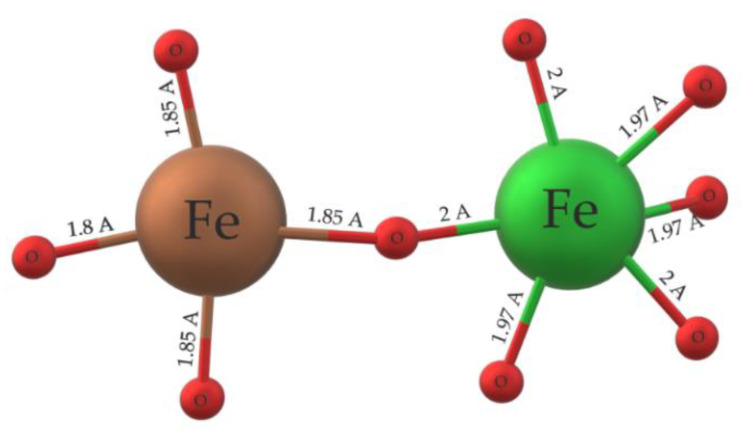
The cell unit of Bi_2_Fe_4_O_9_ [78].

**Figure 25 materials-15-08719-f025:**
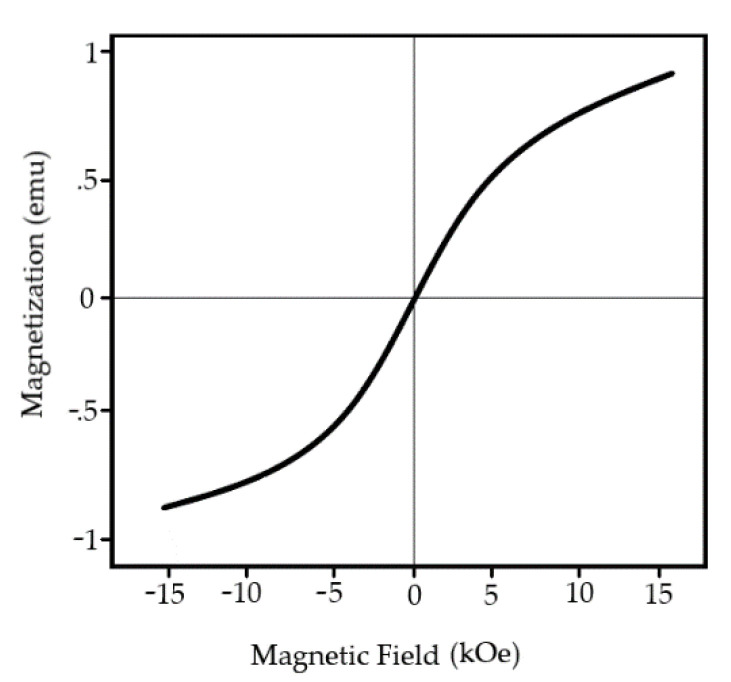
Hysteresis loop of Mullite ferrite material at room temperature.

**Figure 26 materials-15-08719-f026:**
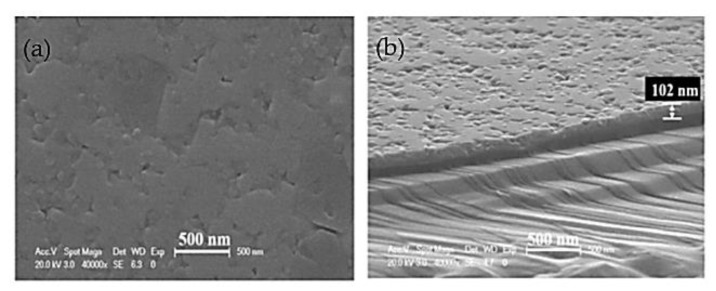
Bi_2_Fe_4_O_9_ film: (**a**) topology; (**b**) cross–section [82].

**Figure 27 materials-15-08719-f027:**
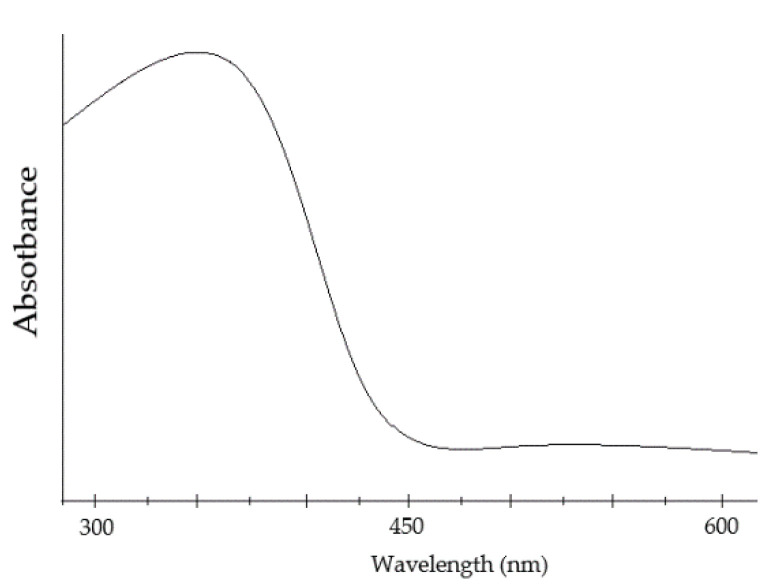
Light absorption curve of Bi_2_Fe_4_O_9_.

**Figure 28 materials-15-08719-f028:**
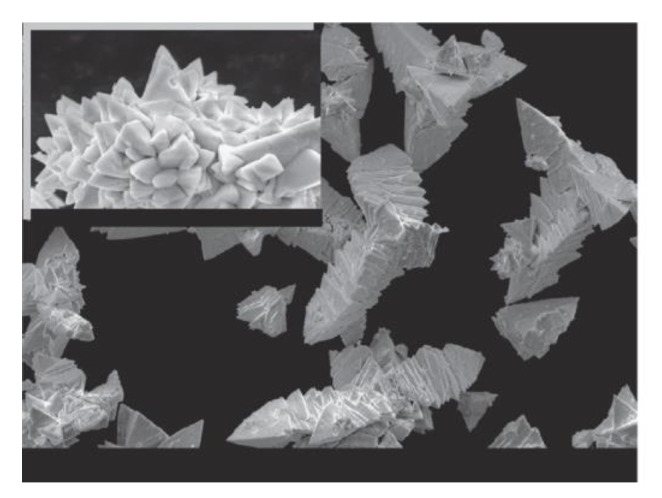
Iron Selenite powder crystals [91].

**Figure 29 materials-15-08719-f029:**
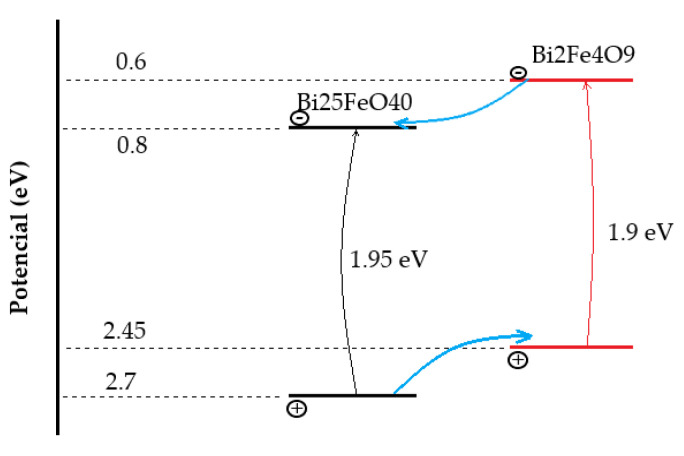
Photocatalytic [98] reaction mechanism of iron Selenite and Mullite under light radiation.

**Figure 30 materials-15-08719-f030:**
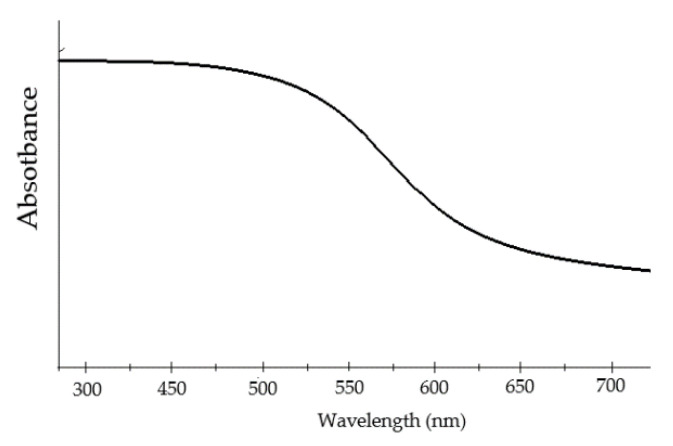
Visible light [47] absorption spectrum [104] of Iron Selenite.

**Table 1 materials-15-08719-t001:** Comparison table of magnetic parameters of pure and doped by Mn BFO material.

Material	Coercivity (Oe)	Magnetization (emu)	Retentivity (emu)
BFO	~70	~0.08	~920 µ
BMnFO–0.1	~280	~0.12	~5.5 m
BMnFO–0.2	~130	~0.17	~3 m

**Table 2 materials-15-08719-t002:** Comparison table of deposition methods.

Synthesis Method	Morphology	Thickness	Cost	Quality
PLD	Thin films, micro/nano crystals, heterostructures.	1 nm–>1 µm	High	High
ALD	Thin films, single layer deposition. micro/nano crystals, heterostructures.	1 nm–>1 µm	High	High
Sol gel method	Thin films	100 nm–>1 µm	Low–medium	Low

## Data Availability

The study did not report any data.

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
