# Peer review of "Brief Theoretical Overview of Bi-Fe-O Based Thin Films"

_materials, 2022, doi:10.3390/ma15248719_

Round 1
Reviewer 1 Report
Misiurev et al., have presented the short review on the title: Brief theoretical overview of Bi-Fe-O based thin films, the topic is as vast as to summarize it in the few pages is so difficult. I think authors have tried their best to make an impactful overview of the topic but that is not enough in my suggestion. I think there are many parameters which must be included in the review article to make it interesting for readers. I suggest rejection for this manuscript and give some suggestions to authors to have a look and consider them if possible for the submission in other journal.
1. Firstly the topic, authors have mentioned Bi-Fe-O, it totally has different meaning than to write Bismuth Ferrite (BiFeO3), please correct this.
2. In the abstract portion, authors have written Bismuth Ferrite directly to BFO without mentioning BiFeO3, and followed the same pattern in the introduction. In fact on the Page 2, line 50 they have first time used the BiFeO3. It’s so strange for me as authors have selected the material for review and they have first time used it in line 50?
3. Superscript and subscript must be revised in units and formula of the material throughout the manuscript.
4. In the start of the introduction portion, authors have described the simultaneously ferroelectric and ferromagnetic properties of BFO, which is specifically termed as “electromagnetic coupling”, and this is the phenomenon especially related with BFO, but in the discussion of 4 paragraph till line 64, I have missed this term.
5. Line 71, please correct the units for polarization.
6. Line 94, please correct the valance states superscripts of Fe and Bi.
7. Most important point about the structure of BFO is its rhombohedral structure, which should be in the start of manuscript but mentioned in the line 84, secondly there is no description of its space group or point group as authors are discussing its lattice parameters.
8. For the Neel temperature discussion of BFO, there must be the discussion of canted G-type structure to elaborate its magnetic behavior.
9. Important: In the introduction portion there is the most basic information of the BFO material, which can be suggested as the initials of BFO. For such high impact factor journal, it should not be acceptable; I think there must be the discussion of market values of BFO as ferroelectric and piezoelectric devices, there must be discussion of ferroelectric analysis for BFO ceramics, thinfilms and single crystals.
10. More the doping factor at the A and B-sites of the BFO material has much importance, researchers have not only studied the A and B site single and co-doping elements but also studied the composite form of BFO like, BiFeO3-xBi0.5K0.5TiO3, BiFeO3-xBi0.5Na0.5TiO3, BiFeO3-xBaTiO3 etc. the discussion is missing.
11. There must be the discussion of morphotropic phase boundary region (MPB) of BFO with its dopants or composite form.
12. There exist the structural transformation depending on the doping elements and also the temperature dependent transformation, the discussion is missing.
13. Section 2.1 is the basic phenomenon of the magnetic hysteresis loop and how it is produced in BFO based material. But still authors have not cited the research articles which have shown the much improved ferromagnetic results. Infact there must be comparison table for this study for the pure BFO and doped and composite form materials of BFO to make it attractive.
14. In the line 170-173, there are 45 references, its so strange in the distance of few words there are so many references?
15. Authors use word BFO and wherever they write BiFeO3 and then they write Bismuth ferrite, please follow a single pattern.
16. Page 5, Line 175“BFO thin films demonstrate different crystallographic orientation bases on ration….” Authos have used this sentence but not mentioned the various orientations of 71o, 109o and 180o, please highlight it.
17. In the Figure 10, schematic, please correct the spelling of silicon.
18. In the discussion of thin film BFO, Photoelectric properties, magnetic properties. I have just studied the basic information of the parameters, I have missed the detailed study of the literature, its values its comparison for dopants or with other perovskites.
19. In section 4, Deposition methods of BFO thin films, authors have just written the techniques and the basic methods of this technique, but they have missed the specific parameters temperature, time, calcination, etc for fabrication.
Author Response
Misiurev et al., have presented the short review on the title: Brief theoretical overview of Bi-Fe-O based thin films, the topic is as vast as to summarize it in the few pages is so difficult. I think authors have tried their best to make an impactful overview of the topic but that is not enough in my suggestion. I think there are many parameters which must be included in the review article to make it interesting for readers.
I suggest rejection for this manuscript and give some suggestions to authors to have a look and consider them if possible for the submission in other journal.
- Firstly the topic, authors have mentioned Bi-Fe-O, it totally has different meaning than to write Bismuth Ferrite (BiFeO3), please correct this.
Bi-Fe-O was corrected
- In the abstract portion, authors have written Bismuth Ferrite directly to BFO without mentioning BiFeO3, and followed the same pattern in the introduction. In fact on the Page 2, line 50 they have first time used the BiFeO3. It’s so strange for me as authors have selected the material for review and they have first time used it in line 50?
Pattern of BFO was corrected and used throughout the manuscript
- Superscript and subscript must be revised in units and formula of the material throughout the manuscript.
Superscript and subscript were corrected
- In the start of the introduction portion, authors have described the simultaneously ferroelectric and ferromagnetic properties of BFO, which is specifically termed as “electromagnetic coupling”, and this is the phenomenon especially related with BFO, but in the discussion of 4 paragraph till line 64, I have missed this term.
The term electromagnetic coupling was added
- Line 71, please correct the units for polarization.
Units of polarization was corrected
- Line 94, please correct the valance states superscripts of Fe and Bi.
Valance was corrected
- Most important point about the structure of BFO is its rhombohedral structure, which should be in the start of manuscript but mentioned in the line 84, secondly there is no description of its space group or point group as authors are discussing its lattice parameters.
The description of space group was provided
- For the Neel temperature discussion of BFO, there must be the discussion of canted G-type structure to elaborate its magnetic behavior.
Neel and Curry temperature was moved to magnetic properties
- Important: In the introduction portion there is the most basic information of the BFO material, which can be suggested as the initials of BFO. For such high impact factor journal, it should not be acceptable; I think there must be the discussion of market values of BFO as ferroelectric and piezoelectric devices, there must be discussion of ferroelectric analysis for BFO ceramics, thin films and single crystals.
Discussion was added
- More the doping factor at the A and B-sites of the BFO material has much importance, researchers have not only studied the A and B site single and co-doping elements but also studied the composite form of BFO like, BiFeO3-xBi5K0.5TiO3, BiFeO3-xBi0.5Na0.5TiO3, BiFeO3-xBaTiO3etc. the discussion is missing.
Discussion was added
- There must be the discussion of morphotropic phase boundary region (MPB) of BFO with its dopants or composite form.
Discussion was added
- There exist the structural transformation depending on the doping elements and also the temperature dependent transformation, the discussion is missing.
Discussion was added
- Section 2.1 is the basic phenomenon of the magnetic hysteresis loop and how it is produced in BFO based material. But still authors have not cited the research articles which have shown the much improved ferromagnetic results. In fact there must be comparison table for this study for the pure BFO and doped and composite form materials of BFO to make it attractive.
The table was added
- In the line 170-173, there are 45 references, its so strange in the distance of few words there are so many references?
References were corrected
- Authors use word BFO and wherever they write BiFeO3 and then they write Bismuth ferrite, please follow a single pattern.
Single pattern was used throughout the manuscript
- Page 5, Line 175“ BFO thin films demonstrate different crystallographic orientation bases on ration….” Authos have used this sentence but not mentioned the various orientations of 71o, 109oand 180o, please highlight it.
Ration angles were highlighted
- In the Figure 10, schematic, please correct the spelling of silicon.
Spelling of silicon was corrected
- In the discussion of thin film BFO, Photoelectric properties, magnetic properties. I have just studied the basic information of the parameters, I have missed the detailed study of the literature, its values its comparison for dopants or with other perovskites.
Main purpose of the manuscript is brief review of BFO material and BFO based thin films. Detail study would be much more relevant for next publication which aim is comparison/influence of impurities on electromagnetic coupling of BFO thin films.
- In section 4, Deposition methods of BFO thin films, authors have just written the techniques and the basic methods of this technique, but they have missed the specific parameters temperature, time, calcination, etc for fabrication.
Parameters of deposition processes were added and highlighted.

Reviewer 2 Report
Authors give a brief review about Bi-Fe-O system and discuss its growth, characterization and application way. The number of paper reviewed is enough but the proposed outline makes me confused. Especially for the title, the discussed scope is more than the type of films. There are some remarkable format errors in the whole manuscript. The subtitle is out of sequence. For instance, where is section 2.2? And the quality of all pictures is far from publishable status. I suggest the authors reorganize the framework and resubmit the draft.
Author Response
Authors give a brief review about Bi-Fe-O system and discuss its growth, characterization and application way. The number of paper reviewed is enough but the proposed outline makes me confused. Especially for the title, the discussed scope is more than the type of films. There are some remarkable format errors in the whole manuscript. The subtitle is out of sequence. For instance, where is section 2.2? And the quality of all pictures is far from publishable status. I suggest the authors reorganize the framework and resubmit the draft.
- There are some remarkable format errors in the whole manuscript.
The manuscript was checked, and mistakes related to format have been fixed
- The number of paper reviewed is enough but the proposed outline makes me confused. Especially for the title, the discussed scope is more than the type of films.
The outline of the review suggests description of BFO material which include properties, structures, perspectives. In addition to BFO material there is overview of BFO based thin films, their deposition methods properties and perspectives.
- The subtitle is out of sequence. For instance, where is section 2.2?
Section 2.2 was added to the manuscript
- The quality of all pictures is far from publishable status.
Overall quality of all pictures was increased.

Reviewer 3 Report
This paper systematically summarizes Bismuth ferrite, mainly discusses its unique ferroelectric properties and magnetism, and analyzes its preparation methods, common forms, development prospects and trends.
For this review article, the author has made a comprehensive understanding of bismuth ferrite materials through reading a large number of documents and his own practical experience. The overall structure of the article is reasonable, the description is clear, and the existing problems are reasonably improved and adjusted. After careful consideration, this paper can be recommended for publication in the journal
Author Response
Thank you for your recommendation.
Reviewer 4 Report
I understand this is a revised version of the manuscript that I reviewed in August. The manuscript has been sufficiently improved according to the comments, and I recommend the publication in its present form.
Author Response
Thank you for your recommendation.

Round 2
Reviewer 1 Report
Misiurev et al. have presented the manuscript titled: Brief theoretical overview of Bi-Fe-O based thin films. Overall presentation of the proposed work is good, but there requiring some modifications which I think are necessary to explain before publication.
1. Page 1, abstract: line 12, “Bismuth ferrite can be produced” please change the word produced by fabricated.
2. Page 1, abstract: line 15, “perspectives and potential applications of the material will be named” change the word named with highlighted.
3. In my previous review, I suggested the authors to discuss the structural evolution in the BFO based material regarding doping, I suggest the authors to check the work especially Figure 7 of “The development of BiFeO3-based ceramics. Chinese science bulletin, 59(36), 5161-5169.”
4. In my previous review I suggested the authors to mention the technological advancement in the BFO based ceramics and thin films in the introduction section please check the recent work on BFO based materials, like “Nano-Micro Letters, 12(1), 1-23.
5. Page 5, Line 155, In the manuscript authors have used the word “Curry Temperature” please correct it with Curie Temperature.
Author Response
Misiurev et al. have presented the manuscript titled: Brief theoretical overview of Bi-Fe-O based thin films. Overall presentation of the proposed work is good, but there requiring some modifications which I think are necessary to explain before publication.
- Page 1, abstract: line 12, “Bismuth ferrite can be produced” please change the word produced by fabricated.
The word produced was substituted by fabricated
- Page 1, abstract: line 15, “perspectives and potential applications of the material will be named” change the word named with highlighted.
The word named was substituted by highlighted
- In my previous review, I suggested the authors to discuss the structural evolution in the BFO based material regarding doping, I suggest the authors to check the work especially Figure 7 of “The development of BiFeO3-based ceramics. Chinese science bulletin, 59(36), 5161-5169.”
Discussion regarding impurities of BFO material was added
- In my previous review I suggested the authors to mention the technological advancement in the BFO based ceramics and thin films in the introduction section please check the recent work on BFO based materials, like “Nano-Micro Letters, 12(1), 1-23.
The introduction section was completely reorganized in order to create and follow defined sequence. Discussion was added
- Page 5, Line 155, In the manuscript authors have used the word “Curry Temperature” please correct it with Curie Temperature.
Curry Temperature was corrected
Reviewer 2 Report
I am satisfied with the revised version and thus recommend it for publication as it is.
Author Response
Thank you for your review